REGISTERED REPORT PROTOCOL

# Research on rapier loom fault system based on cloud-side collaboration

**Yanjun Xiao, Kuan Wang, Weiling Liu, Kai Peng, Feng Wan**  *

College of Mechanical Engineering, Hebei University of Technology, Tianjin, China

* wf_hebut@163.com

This is a Registered Report and may have an associated publication; please check the article page on the journal site for any related articles.

## Abstract

The electrical control system of rapier weaving machines is susceptible to various disturbances during operation and is prone to failures. This will seriously affect the production and a fault diagnosis system is needed to reduce this effect. However, the existing popular fault diagnosis systems and methods need to be improved due to the limitations of rapier weaving machine process and electrical characteristics. Based on this, this paper presents an in-depth study of rapier loom fault diagnosis system and proposes a rapier loom fault diagnosis method combining edge expert system and cloud-based rough set and Bayesian network. By analyzing the process and fault characteristics of rapier loom, the electrical faults of rapier loom are classified into common faults and other faults according to the frequency of occurrence. An expert system is built in the field for edge computing based on knowledge fault diagnosis experience to diagnose common loom faults and reduce the computing pressure in the cloud. Collect loom fault data in the cloud, train loom fault diagnosis algorithms to diagnose other faults, and handle other faults diagnosed by the expert system. The effectiveness of loom fault diagnosis is verified by on-site operation and remote monitoring of the loom human-machine interaction system. Technical examples are provided for the research of loom fault diagnosis system.

## Introduction

Fault diagnosis techniques for equipment have prominent applications in electric power systems [1, 2], chemical process systems [3, 4], photovoltaic systems [5–9], bearings [10, 11], building energy systems [6], control systems [12, 13], and automation equipment [14]. The electrical control system of rapier loom is susceptible to various disturbances during operation and is prone to malfunction. This will seriously affect the production and a fault diagnosis system is required to reduce this effect.

Due to the lack of sufficient computing and storage resources for weaving machine equipment, weaving machine fault diagnosis systems need to process fault data in the cloud using IoT technology. However, with the development of IoT technology, the number of connected devices has increased, operational data has increased significantly, and cloud-based applications are expanding [15]. Improving the efficiency of data processing resources in the cloud to

**Data Availability Statement:** All relevant data are within the paper and its Supporting information files.

**Funding:** This study was supported through an award from the fifth "333 Project" training fund project of Jiangsu Province issued by the Department of Human Resources and Social Security of Jiangsu Province: Research on Weaving Machine Decentralized Self-organizing Network and Remote Fault Diagnosis Expert System (Project No. BRA2020244). The funders had no role in study design, data collection, and analysis, decision to publish, or preparation of the manuscript.

**Competing interests:** XY is the chairman and founder of Jiangsu Corey Intelligent Control Automation Technology Co. but does not receive a salary from them. This does not alter our adherence to PLOS ONE policies on sharing data and materials. There are no patents, products in development or marketed products associated with this research to declare.

reduce consumption has become an important need [16]. The loom fault diagnosis system needs to take this need into account while achieving accurate diagnosis of loom faults.

Many popular fault diagnosis methods exist. Based on big data analysis and processing, [1] proposed a hybrid data mining method based on clustering, association rules and stochastic gradient descent for the classification and prediction of faults in power systems. However, [1] applied to looms lacks sufficient sample data and requires a large amount of computational resources [10]. proposed a cloud/edge collaborative fault diagnosis solution with real-time responsiveness to improve the accuracy of the algorithm and reduce the time cost by improving the transfer learning and edge collaboration of the convolutional neural network model with a small number of samples for deep learning. However, the fault diagnosis approach of [10] is only applicable to faults in bearings and does not consider the electrical characteristics of the loom. The adopted algorithms and computational tasks divided between cloud and edge cannot be applied to loom fault diagnosis. In [12], a fault diagnosis method based on log analysis is used and a clustering algorithm-based system fault diagnosis method is proposed to improve the effect of log clustering and to improve the accuracy of fault diagnosis while ensuring the efficiency of fault diagnosis. However, the method used by [12] requires a large amount of storage and computational resources for saving and processing logs, which requires more resources for weaving machine systems with more models. [14] proposed the main strategies for fault prevention and maintenance methods for common faults in electrical automation equipment, and combined field fault information collection, fault information fusion and information linking techniques to improve the accuracy of fault diagnosis for the problem of low number of correctly detected fault parameters. However, effective machine learning methods to improve the accuracy of fault diagnosis are lacking, and the need to reduce the pressure of cloud computing is not considered.

Although the above fault diagnosis methods have their own advantages, they do not consider the electrical characteristics of the loom and cannot be directly applied to the loom. In addition, the above fault diagnosis methods mainly focus on how to improve the accuracy of fault diagnosis. No research has been conducted for the exponential growth of data in industrial IoT to improve the efficiency of data processing in the cloud to reduce the pressure of cloud computing.

The above fault diagnosis methods are limited by the process and electrical characteristics of the weaving machine and are deficient. The method of method [1] lacks sufficient sample data. The method of [10] is of value, but this fault diagnosis method for bearings is not applicable to diagnose electrical faults of weaving machines. The method of [12] lacks sufficient computational and storage resources for weaving machine systems. The method of [7] is not applicable to all weaving machine fault cases. The method of [14] is informative but lacks effective machine learning methods to improve the accuracy of fault diagnosis. The existing popular fault diagnosis methods need to be improved.

Previous studies on loom monitoring and fault diagnosis systems have been carried out with some results. [17] and [18] applied IoT technology to loom systems. [19] improved the loom productivity by studying loom fault diagnosis methods. [20–22] studied loom monitoring system regarding loom working parameters, condition monitoring key technologies, reliability and maintainability respectively to achieve effective monitoring of loom operation status. [23–25] studied the loom fault diagnosis methods, advanced loom technologies and common electrical faults, and the fault diagnosis methods used are worthy of reference.

However, these troubleshooting approaches do not consider the data and computational resources required to perform troubleshooting in the cloud. Many existing studies on cloud computing are worthy of consideration [26]. Integrating cognition into computing systems to achieve accurate prediction of impending failures. [27] improves the efficiency of IoT by

adding cloud infrastructure [28]. Improved infrastructure data transfer performance to identify anomalies in sensor data [29]. Improved security of mobile cloud computing using modular encryption standards. In the future trend of big data explosion, significant increase in the number of devices access and shortage of data computing resources in cloud platforms, there is a need to improve the existing loom fault diagnosis methods to reduce the pressure on cloud computing [15]. Edge computing is expected to solve this problem [30]. However, it needs to be studied according to the process and electrical characteristics needs of rapier looms.

We researched the early stage and designed a special-machined rapier control system for the hardware experiment platform. In [17] we researched the Internet of Things technology, designed the loom remote monitoring system. In [23] we studied the fault diagnosis algorithm and proposed a loom fault diagnosis method based on rough set and Bayesian network. However, the rapier loom fault diagnosis method used in previous studies requires a large amount of computing and network resources on the cloud platform, and the cost is relatively high. This will limit the large-scale commercial use of this method, and edge computing technology needs to be introduced to solve this problem.

In order to meet the needs of rapier weaving machine process and electrical characteristics, this paper is an in-depth study on the previous research. A rapier loom fault diagnosis method combining edge expert system and cloud-based rough set and Bayesian network is proposed. Based on the experience of field commissioning and maintenance experts, the electrical faults of rapier looms are classified into common faults and other faults according to the frequency of occurrence, and the causes of common faults are listed. Based on the knowledge of maintenance experts for common faults a knowledge base is constructed in the field based on knowledge fault diagnosis experience to build an expert system for edge computing to diagnose common loom faults and reduce the pressure of cloud computing. Build a cloud fault diagnosis system using loom fault diagnosis method based on rough set and Bayesian network. Collect loom fault data in the cloud, train loom fault diagnosis algorithms to diagnose other faults, and handle other faults diagnosed by the expert system. A rapier loom weaving machine human-machine interaction system is built as an experimental platform to verify the fault diagnosis effect of the edge expert system and the cloud fault diagnosis system. A technical example of a loom fault diagnosis system based on cloud-edge collaboration is provided. The resource allocation and task offloading of the edge and cloud are reasonably allocated according to the electrical characteristics and fault conditions of the loom, thus reducing the pressure on cloud computing.

The rest of the paper is structured as follows. Chapter 2 analyzes the process and common faults of rapier looms. Chapter 3 investigates the expert system based fault system of rapier loom. Chapter 4 investigates the fault diagnosis system based on rough set and Bayesian network in the cloud. Chapter 5 introduces the experimental platform and fault diagnosis effect of this fault diagnosis system. Chapter 6 is the summary of the whole text.

## Materials and methods

### Data preparation

Previous studies have provided some data. [17] designed a loom remote monitoring and fault diagnosis method that provides data on loom remote monitoring and data on the corresponding time for loom fault diagnosis. It is shown that the loom remote monitoring system can meet the demand for remote monitoring of looms and the response time of fault diagnosis meets the industrial demand. A loom fault diagnosis method based on rough sets and Bayesian networks is investigated in [23], which provides the loom fault diagnosis method and example data. The example data illustrate how this fault diagnosis method enables fault diagnosis of

weaving machines. [23] also provides data comparing the fault diagnosis effectiveness of this method with other methods to show that its fault diagnosis is more accurate than other methods. Table 1 presents a summary of these datasets.

## Rapier loom main weaving process and fault analysis

This chapter analyzes and studies the main process principles and fault types of rapier loom weaving and their solutions from the structural composition of each module of the whole control system.

**Analysis of the main process principle.** The operation of the loom is divided into two types: debugging and maintenance and automatic production. In the debugging mode, the operator can debug the loom-related movements in a single step as needed, and can also query the diagnosis results in case of loom failure; while in the automatic mode is the normal automatic production state.

The HMI interface of the field system starts with a login page, which is used to enter the HMI field system according to the mode selection and user rights password. In the maintenance and debugging mode, maintenance workers or engineers can directly enter the maintenance and debugging interface to perform relevant operations. The main functions include: single-step operation of the main weaving machine processes, real-time monitoring of the warp and weft tension curves, and downloading of tension data during the manufacturing process, signal shielding, and restoration of factory settings.

In automatic mode, after a blocker or other user logs into the system, the first default display is the status desktop, which mainly realizes the loom running speed, shift information, pattern color information, the current action of the loom, the running time and the alarm window. In addition also as the desktop can jump to the parameter settings, color editing, signal monitoring, alarm log, auxiliary center and other functional interfaces for operation.

Before the automatic operation, the yarn layout should be completed according to the requirements, and then the tension adjustment is set through the tight yarn or tight roll operation of the HMI on-site system. The on-site HMI interface can provide real-time feedback of the tension sensor detection data for the staff to check whether the fabric tension meets the requirements, and then the tension is locked after the requirements are met. Then press the automatic operation button, the loom starts to execute the process of opening, weft selection, weft leading, weft stranding and weft beating in turn. In this process, as long as there is no malfunction or stop command, the loom always weaves at the set running speed, in which the warp feeder and the winder continuously transport the yarn and wind the fabric, while the other mechanisms perform the corresponding actions according to the spindle rotation angle. Fig 1 shows the overall process flow.

During the normal weaving process of a rapier weaving machine, assuming that the rapier weaving machine starts at a spindle angle of 320˚, it must be ensured that the weft selection finger must pull the weft yarn and extend it into place before the weft guide rapier arrives, and

**Table 1. Datasets from existing studies for loom fault diagnosis methods.**

| Source | Description | Features |
|---|---|---|
| [17] | Loom remote monitoring interface and monitoring effect | Remote monitoring of weaving machines is possible |
| | Loom fault diagnosis response time | Response time to meet industrial demands |
| [23] | Loom troubleshooting methods and examples | Enables fault diagnosis of weaving machines |
| | Loom fault diagnosis effect | Better fault diagnosis accuracy than other methods |

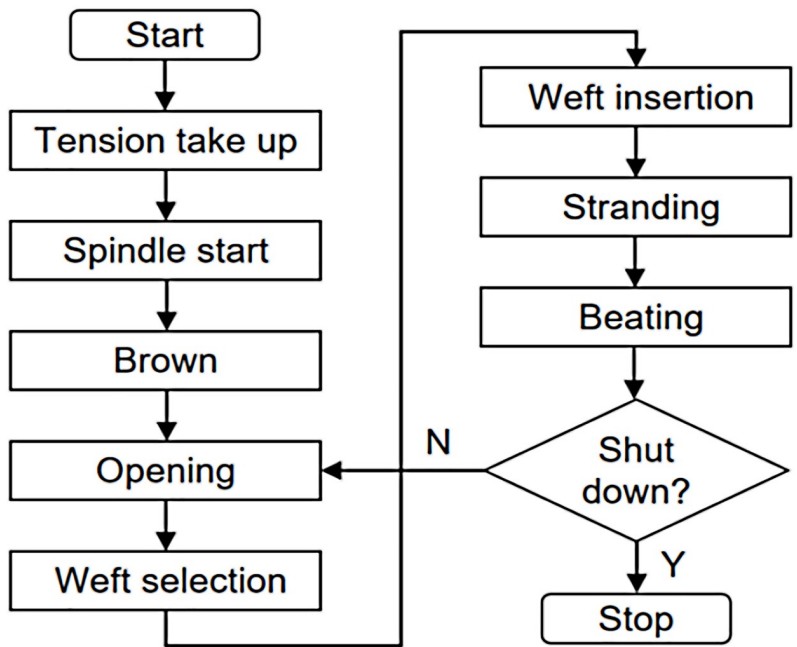

**Fig 1. Overall process flow.**

only then can the rapier entrain the weft yarn into the bobbin. The rapier weaving machine remains stationary for a certain period of time during the maximum opening, when the weft picker finger starts to retract while the rapier belt is crossing the bobbin opening; the control system then gives a switching signal to the electronic dobby to start the switching of the brown frame, and the weft beating process starts when the rapier belt with the weft yarn leaves the bobbin opening, while the edge-stranding mechanism completes the edge-stranding process and then enters the next opening process again. The rapier weaving machine completes the weaving work in this process cycle. Because of the need to control multiple mechanisms to complete multiple processes, the failure situation is complex and the cause of the failure is difficult to determine.

**Fault analysis of rapier loom.** The faults that occur during the operation of rapier looms can be divided into mechanical and electrical faults. Since the mechanical structure is rarely faulty and difficult to monitor, this system only analyzes the electrical faults that occur in rapier looms.

According to the different causes of failure, the failure is divided into electrical failure, sensor failure and control device failure. The electrical appliances include servo driver, control circuit board, inverter, transformer, switching power supply, etc.; sensors include encoder, proximity switch, warp break sensor, weft detector, tension sensor, oil level, oil pressure sensor; control devices include main clutch, slow clutch, brake, weft clutch, main motor, slow motor, electronic dobby, warp feeding and winding servo motor, fan, weft storage device.

One or more reasons may be responsible for a failure of a rapier loom. For example, excessive tension fluctuations during the weaving process of the loom cause uneven yarn density distribution on the fabric surface. Setting tension value error, warp beam diameter parameter setting error, PID algorithm parameter error, tension sensor failure, feeding warp winding servo motor failure are able to cause excessive tension fault phenomenon, so through the debugging experience summed up eighteen common fault information and the corresponding

fault relief methods, Tables 2 and 3 for the rapier loom main fault and its maintenance methods.

Through the above experience summary of rapier loom fault types and solutions it can be observed that a fault often corresponds to multiple possible causes of failure. The rapier loom fault diagnosis system needs to accurately give the key part or cause of the fault and its solution to the maintenance workers through the human-machine interaction interface when the equipment fails. Due to the complexity of loom fault situations, there are other fault situations in addition to the above common fault situations where the cause of the fault is difficult to determine. Loom faults can be divided into common faults and other faults. Common faults occur more frequently and have more fault data samples, making it easy to obtain the knowledge needed to perform fault diagnosis. Other faults occur relatively infrequently and have few fault data samples, and need to be processed using fault diagnosis algorithms. The fault diagnosis algorithm must have sufficient data samples and computational resources to deal with loom fault problems. Due to the limited computing and storage resources of on-site equipment, the cloud platform must be used for fault diagnosis of loom equipment. Because looms do not fail in real time, the cloud platform of loom fault diagnosis system has limited computing resources for fault diagnosis. When the number of looms in the system is large, the simultaneous occurrence of multiple loom failures will put enormous computational pressure on the cloud platform.

In order to reduce the pressure of cloud computing for loom fault diagnosis system, a cloud-edge collaborative fault diagnosis method needs to be investigated. An expert system is constructed using the resources of the human-machine interaction system of the field loom to handle known electrical faults by edge computing. These electrical faults can be solved using knowledge-based fault diagnosis experience due to their high frequency of occurrence. For more complex cases, no explicit knowledge is available to handle such cases due to their low frequency of occurrence and small data samples, and cloud computing is needed for processing.

**Table 2. Common repair methods.**

| Number | Troubleshooting |
|--------|-----------------|
| T1 | Check the oil circuit to determine if refueling is needed |
| T2 | Reset the motor thermal protection relay and check |
| T3 | Whether the motor is blocked by foreign objects or inflexible operation |
| T4 | Check if the main motor thermal protection relay protection current is set too small |
| T5 | Timely cleaning of fans |
| T6 | Check fan-related circuits |
| T7 | Check if the weft storage is normal |
| T8 | Weft detector failure or adjustment of the sensitivity of weft detection |
| T9 | Weft selection finger is not in place |
| T10 | Encoder failure |
| T11 | Consult the servo driver manual according to the code displayed on the servo motor driver in the electronic control box |
|  | According to the instructions to solve the servo motor alarm requires re-powering to restore |
| T12 | Check whether the upper and lower limits of the set tension value are set reasonably |
| T13 | Check the warp beam diameter setting for accuracy |
| T14 | Check if the warp feeding and winding servo motor is normal |
| T15 | Check PID parameter settings |
| T16 | Check the tension sensor wiring |

**Table 3. Common faults of rapier looms and their repair methods.**

| Common faults | Troubleshooting number |
|---|---|
| Oil pressure level is too low | T1 |
| Overload of main motor and slow motor | T2, T3, T4 |
| Fan overload | T5, T6 |
| Broken weft or double weft | T7, T8, T9, T10 |
| Warp feeding and winding motor failure | T11 |
| Warp tension overrun | T12, T13, T14, T15, T16 |
| Encoder failure | T17, T18, T19 |
| Main clutch failure | T20, T21, T22 |
| Brake, latitude clutch, slow clutch failure | T23, T24, T22 |
| Proximity switch does not light up | T25, T26 |
| Loom positioning timeout | t27, t28, t29, t30, t31, t32 |
| Excessive braking distance | T33, T34, T35 |
| Weaving machine stop timeout | T36 |

## Fault diagnosis method research

**Expert system-based fault diagnosis method.** Expert systems usually consist of six parts: knowledge base, reasoning machine, interpreter, database, knowledge acquisition, and human-computer interaction interface. The main role of the knowledge base in an expert system is the storage of various kinds of knowledge. There are various expressions of knowledge bases, including knowledge generative rule representation, knowledge semantic network representation, and knowledge framework representation. Among them, the knowledge generative rule representation is expressed in the format of.

$$IF\langle Condition\ 1\rangle\langle Condition\ 2\rangle\cdots\langle Condition\ N\rangle THEN\langle Conclusion\rangle$$

This representation is simple in logic and easy to understand, simplifies the reasoning and calculation process, and meets the shortcomings of MCGS Pro configuration software which is not strong in calculation, but the representation is prone to reasoning contradictions, and attention should be paid to its consistency and logic in the establishment process.

The knowledge semantic network representation has the disadvantages of difficulty in constructing knowledge and unclear retrieval of expression results, and its semantic network structure is difficult to express the nodal events and relational arcs in it by MCGS Pro logic scripts. And the knowledge frame representation uses frames to express the logical details of the diagnosis process, so the process knowledge experience is difficult to express logically, so this system selects the knowledge generative rule representation as the expression of the knowledge base.

The database is mainly used for the storage of intermediate data generated by the expert system in the process of dynamic changes and the inference conclusions generated by the reasoning machine. The database is capable of both pushing inference conclusions and explanatory notes to the interpreter and providing storage space to the reasoning machine.

The reasoning machine is the brain of the expert system, which reasoned about the diagnosis results according to the set methodological strategy through the expert knowledge stored in the knowledge base. The reasoning process of a reasoning machine is usually independent of the content of the knowledge and is only related to the logical method of expressing the knowledge.

The HMI is a bridge between the user, the designer, the engineer and the expert system, and the engineer or designer uses the HMI as a platform to improve the knowledge representation in the knowledge base. Reasoners and interpreters also use the HMI as a platform to present diagnostic results and fault explanations to users. The existence of the HMI allows the user to intuitively understand the causes of faults and repair methods, making it easier to solve fault problems.

Most of the traditional rapier loom HMI systems come with fault alarm functions, while lacking fault diagnosis functions, thus realizing maintenance assistance to on-site maintenance workers. For the rapier loom faults in the experimental platform, a fault diagnosis expert system is added.

In the normal operation of the system, each sensing and detection signal is used as the input of the control system for the control of the system, and the system will also produce related output and intermediate quantities, these signals are not all expressed individually they have a certain logical relationship, a fault in the system means that a certain signal quantity is abnormal, so the logical relationship related to the signal quantity will be destroyed. This logical relationship will be combined with the actual fault expert's experience to reason about the occurrence of the fault based on the knowledge base representation, and finally arrive at the fault diagnosis results.

Taking the surface fault of weft breakage as an example, four equipment faults, namely encoder, weft storage, weft detection and weft selection finger, can be the cause of weft breakage on the loom. The encoder failure will cause the system to read the spindle angle incorrectly, which will lead to the weft detection angle range or the weft selection finger retracting incorrectly, which will lead to the weft breakage phenomenon. A malfunction of the weft accumulator can directly cause the weft output to break. Weft detection malfunction or inaccurate sensitivity will also cause the system to alarm. Inadequate retraction of the weft selection finger can also cause the sword belt to miss the weft yarn, resulting in broken weft.

In addition the rapier loom control system also includes frequency converters and servo drives, these intelligent devices determine the key to the normal operation of the loom spindle movement and the winding and warp feeding process, and their own fault diagnosis function within the device is also a weakened form of the expert system as a theoretical basis, so the fault library of the intelligent devices as part of the overall knowledge base, using the real-time MCGS Pro configuration software database and scripting logic reasoning ability of MCGS Pro configuration software, and then match the corresponding human-computer interaction interface to realize the fault diagnosis function of the human-computer interaction field system for intelligent equipment.

The rapier loom winding and unwinding mechanism is designed to meet the stability of fabric tension during weaving by precisely controlling the movement of the winding and unwinding rolls with servo motors, and therefore requires the use of servo drives to complete the driving of the servo motors. Also through communication with the control system, the control system can not only read the working parameters of the servo motor, but also collect fault information in case of motor failure. After the fault information sent by the servo drive, the control system processes the fault message frame to obtain the fault alarm number, and the obtained fault alarm number corresponds to the fault variable object set in the real-time database of MCGS Pro configuration software, and then the same internal knowledge is built up in the field system using the representation of IF < condition > THEN < conclusion >. knowledge base.

**Fault diagnosis method based on rough set and Bayesian network.** Rough set is an uncertainty information processing method proposed by Z. Pawlak in 1982. Rough set theory is a theory for analyzing and processing data that deals with uncertain and incomplete data

information. Its core is to reduce the data first, then analyze and process it, and finally obtain a solution. The system uses rough set theory to classify fault information. The loom fault information system is established, the boundaries are set, the knowledge is approximately reduced using decision tables, and the recursive algorithm is applied to find the minimum reduction to obtain the final classification rules, make the corresponding rule judgments for the actual problem, and draw conclusions.

Bayesian networks include directed non-return graphs and local conditional probability distribution tables, which indicate the correlation between nodes and the probability of node correlation. Then there is the formula BN = (G, P), BN is the abbreviation of Bayesian network and G is a directed non-return graph consisting of many nodes indicating the conditional probabilities of the nodes. The Bayesian probability distribution formula is as follows:

$$P(X) = \prod_{i=1}^{n} (X_i | X_{i-1}, \cdots X_1) \tag{1}$$

According to the Bayesian network structure in Fig 2, if the child nodes connected with the parent node are independent of each other, there are:

$$P(X_1, X_2, X_3, X_4, X_5) = P(X_1|X_5) \times P(X_2|X_1, X_5) \times P(X_3|X_2) \times P(X_4|X_5, X_2, X_3) \times P(X_5) \tag{2}$$

For the fault diagnosis of the loom, follow the steps below to construct a Bayesian network.

1. All failure causes and failure results of the loom are represented as network nodes.

2. Connecting the cause and effect of the fault with wires.

3. Find the probability of each node based on the degree of influence of the cause of the failure on the outcome of the failure.

Assume that C is used to denote the fault outcome variable and that C can take $c_1$, $c_2$, ..., $c_n$. If the failure cause variable Xi can take more than one value, it is denoted by xi. In calculating the posterior probability of the system, since the causes of the system failure are known $x_1$, $x_2$, ..., $x_n$, the corresponding probability of the failure outcome ci is therefore

$$P(c_i|x_1, x_2, \ldots x_n) = \frac{P(x_1, x_2, \ldots x_n|c_i) \bullet P(c_i)}{P(x_1, x_2, \ldots x_n)} = \beta \bullet P(x_1, x_2, \ldots x_n|c_i) \bullet P(c_i) \tag{3}$$

β is a fixed value. Because sample data only, a specific Bayesian network structure cannot be determined, so it is also necessary to learn the Bayesian structure. In the case of data samples only, the K2 scoring method can learn the Bayesian network structure. The best K2 scoring method is used for fault diagnosis system. In network learning, it is more difficult to obtain a suitable network structure when the data is incomplete than when the data is complete. This fault diagnosis system will learn the network when the fault data is incomplete. The failure probability is calculated by Bayesian classifiers (e.g., plain Bayesian classifier, TAN Bayesian classifier, and regular Bayesian classifier). The maximum failure result is the final diagnosis. When using the Nb (plain Bayesian classifier) model to calculate the probability of loom failure, the total probability of each failure in the sample is first calculated. The probability of

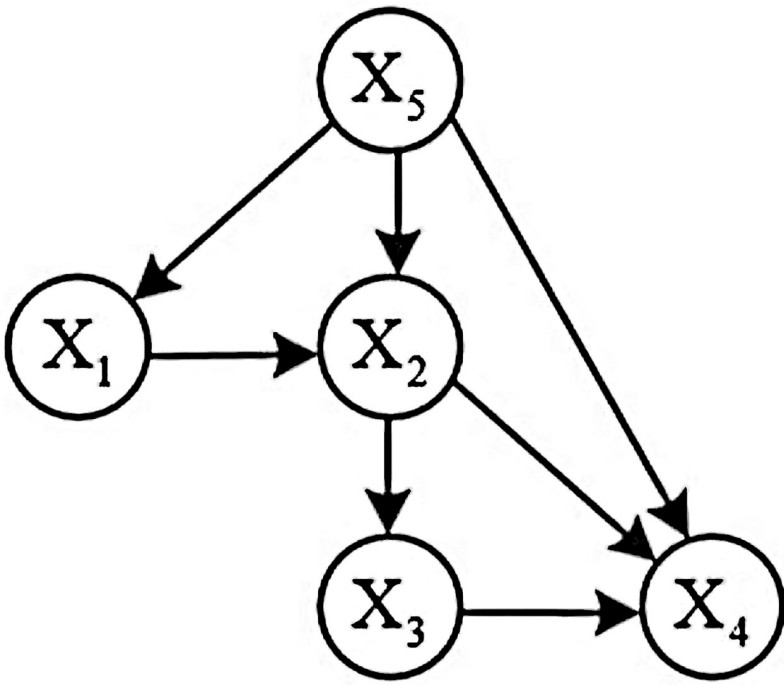

**Fig 2. Overall process flow.**

failure a priori can be calculated by the frequency of failure of.

$$P(C_j) = \frac{N_{cj}}{N} \tag{4}$$

$$P(Y = x_i | C_j) = \frac{N_{cj}^{(xi)}}{N_{cj}} \tag{5}$$

where, $N_{cj}$ is the frequency of the fault outcome and the cause of the fault is xi, $N_{cj}$ is the frequency of the fault outcome. When $N_{cj}^{(xi)} = 0$, the following equation calculates the conditional probability of a high-speed loom failure.

$$P(Y = x_i | C_j) = \frac{1/N}{N_{cj} + N_{xj}/N} \tag{6}$$

By the above method, the probability of all fault outcomes in the sample is calculated and the final diagnosis is found to be the most likely fault outcome.

The loom fault diagnosis method based on rough set and Bayesian network used in this paper is shown in Fig 3. The loom fault detection is achieved by judging whether the data collected by the loom sensors are abnormal or not. The equipment operation parameters collected by the loom's sensors are categorized by the expert system as fault results, and the potential fault causes of the fault results are listed. Applying rough set theory, conditional attributes are determined from the set of fault results and decision attributes are determined from the set of fault causes. A decision table is established based on the causal relationship between the fault cause and the fault result, and the final decision table is obtained by applying the minimum simplification of the decision table by rough set theory. A Bayesian-based network model is

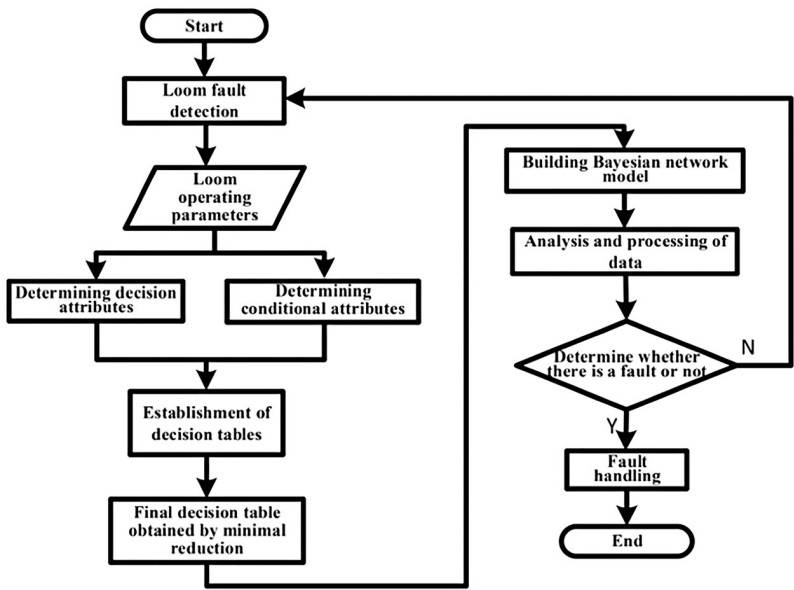

**Fig 3. Block diagram of the fault diagnosis process.**

constructed based on the relationship between the fault cause and the fault result in the decision table. Through Bayesian network, each failure cause is calculated, and the failure cause with the maximum probability is the failure result.

We tested the fault diagnosis method using 200 samples of ten common faults collected in the field, and the fault diagnosis effect of the present method and the fault tree and neural network methods is shown in Fig 4. The fault diagnosis result of fault tree has a higher correct rate for some fault problems but a lower correct rate for others. Among them, the correct rate of diagnosis for 5 kinds of fault problems can be close to 90%, and the correct rate of diagnosis for 4 kinds of fault problems is less than 70%. The correct rate for 6 types of fault problems is similar to this method, while the other 4 types are significantly lower than this method. The correct rate of fault diagnosis fluctuates greatly. The correct rate of the neural network fault diagnosis results for individual fault problems is high, but the correct rate for the 6 fault problems is the lowest. Among them, the correct rate of diagnosis fluctuated in the range of 67%-33% for the six fault problems with a small number of data samples. The neural network is less effective in diagnosing these fault problems. The correct rate of fault diagnosis of the present method is significantly higher than that of the fault tree and neural network methods. Affected by the number of samples of the fault problem, the correct rate of the diagnosis of the present method fluctuates greatly, for example, the correct rate of the diagnosis of the present method is only 80% and 66.70% for the tree samples of winding mechanism fault and inverter fault are too small. However, for the fault problems with more data samples the correct rate of this method is in the range of 87.5%-93.8% with less fluctuation. For example, weft storage failure, warp feed servo failure, winding servo failure and fault exclusion sensor failure. On the contrary, the correct rate of the fault tree and neural network methods for diagnosing different fault problems fluctuates more. Therefore, this method is superior to the fault tree and neural network methods.

## Results and discussion

In order to verify the fault diagnosis effect of this method, a rapier loom human-computer interaction system is built as an experimental platform. The effectiveness of the expert system

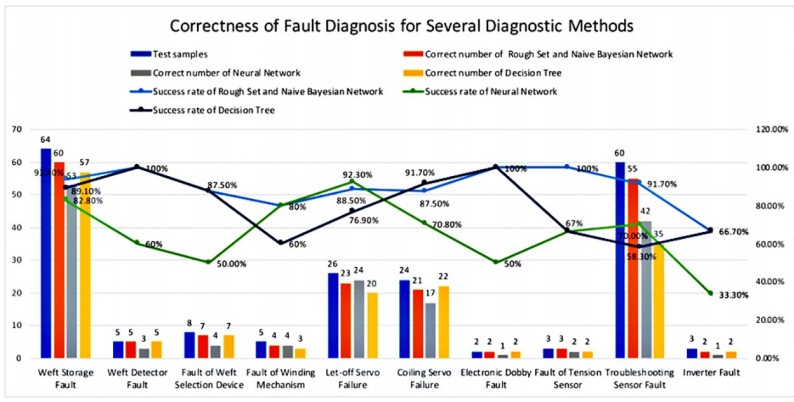

**Fig 4. Comparison of the effect of fault diagnosis methods.**

in diagnosing common faults of rapier looms is verified in the field, and the effectiveness of the diagnosis of other faults of rapier looms is verified by using the remote monitoring system.

## Human-machine interaction system for rapier weaving machines

Based on the analysis of the overall functional requirements of the system, the entire rapier loom HCI system is divided into two parts: on-site HCI and remote HCI.

The human-computer interaction system is based on the touch screen. At present, the domestic touch screen configuration technology is developing rapidly, inexpensive and full-featured to meet the design needs, so this topic selects TPC7072EN type touch screen to carry the field human-computer interaction interface. HMI field system and rapier loom control system data communication, the rapier loom operating parameters, production data and operating instructions and other information are all displayed in the HMI interface, the final workers through the field system human-machine interface to achieve human-machine interaction with a single rapier loom. At present, the rapier loom control system mostly adopts microcontroller as the control core, and the master control system and the field human-machine interaction system realize the data transmission at the physical medium layer and link layer through the industrial bus, and many industrial devices mostly adopt Modbus communication protocol.

In addition, in order to be able to achieve an expert system for fault diagnosis in the field, according to the touch screen configuration comes with a real-time database and a strategy script with strong logic and weak computational power, the fault tree diagnosis method of fast and direct logical relationship clearing is chosen in the diagnosis method, which establishes all the information related to a single apparent fault in a causal relationship with each other, and the fault is analyzed layer by layer to diagnose the root cause of the fault.

The HMI remote system completes the data transmission chain between the loom equipment at the industrial site, the cloud platform at the network and the WEB client. The remote system can be abstractly divided into three levels: the perception layer, the network and the application layer. The sensing layer transmits data from rapier loom working parameters and control commands to the network layer through communication devices, and finally realizes the physical connection between the application layer and the network layer, and then the network layer uploads the data to the Internet and sends it to the cloud platform. The user then interacts with the cloud platform server through monitoring, fault query and other related operations in the WEB client, thus completing the remote human-machine interaction mode

of the rapier loom human-machine interaction system. The communication equipment can be WIFI module, ZIGBEE module or Bluetooth module, etc. The network layer mainly plays the role of transmitting relevant information to achieve data interaction, and needs to meet the security of information transmission, which mainly includes local area network, Internet, carrier network, etc.

Fig 5 shows the overall framework of the HCI system. The rapier loom HCI system is divided into two interaction modes: on-site and remote, which mainly includes: embedded control system, wireless communication module, on-site MCGS man-machine system, cloud platform server, WEB client, etc. The specific description of the system is as follows.

1. In the control system of the master module chip STM32F407ZET6. through the master module on the chip serial port design RS485 communication module hardware circuit, RS485 communication module and the master chip serial port for data transmission, and the touch screen device window using ModbusRTU communication protocol to achieve the data interaction, based on ModbusRTU communication protocol to complete Master module slave driver.

2. Analyze the specific functional tasks of the field system and complete the HMI planning of the field subsystem according to the functional tasks. The human-machine interface is based on the touch screen, and the MCGS Pro configuration software is used to complete the design of the main control window, device window, user window, real-time database, and strategy script, so that the field subsystem can realize the real-time monitoring, parameter setting, pattern editing, fault diagnosis and alarm, and machine repair and debugging of the rapier loom. Among them, the fault diagnosis function adopts the fault tree fault diagnosis method based on the principle of expert system theory, takes the sensor collection signal and the self-contained knowledge base of intelligent devices in the control system as the relevant logical conditions for fault diagnosis, and establishes the expert knowledge base by completing the knowledge expression through the strategy script to realize the fast diagnosis function of field faults.

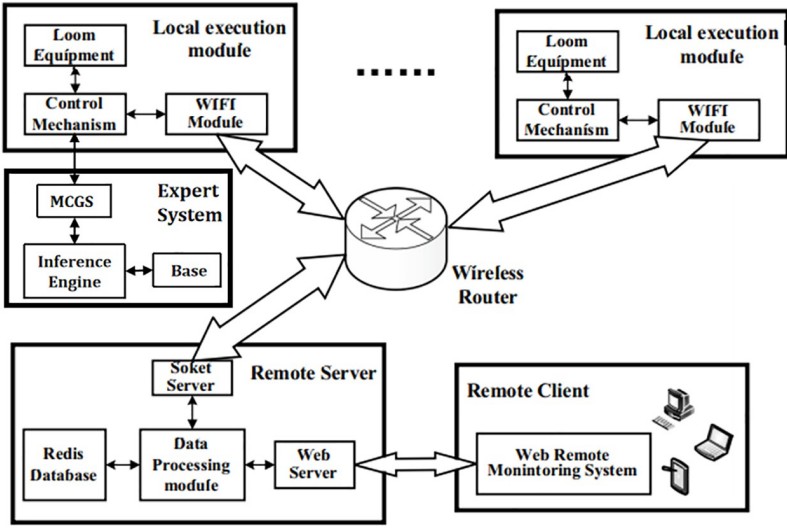

**Fig 5. Overall framework of HCI system.**

3. Between the remote system site control terminal and the cloud platform, the wireless module external circuit is mounted through the serial port of the chip on the master control module, and the corresponding driver design and development is completed to realize the wireless connection between the site control system and the remote subsystem. The master control chip will pass the collected data to the wireless module through the serial port, and the data transmission with the cloud platform server adopts TCP/IP communication protocol based on the socket interface.

4. The cloud platform server is the core mechanism of the rapier loom human-machine interaction remote system, which mainly includes the data processing module WEB server and Redis database, and related service programs. the socket interface to the implemented TCP server and TCP client for data interaction, and the incoming and outgoing transmission data files to the data processing module. The data processing module stores the data files in the Redis database as historical parameters and then processes the device operating parameters and passes the processed parameters to the WEB server. the WEB server mounts an HTML page that feeds the loom operating parameters to the WEB client via Ajax, which enables real-time human-machine interaction with the client for fast refreshing of the device data.

## Rapier weaving machine human-machine interaction system fault diagnosis experiment

The expert system theory is applied to the fault diagnosis, and the abnormal signals are reasoned with a reasonable reasoning mechanism to realize the expert system fault diagnosis function of the rapier loom human-machine interaction field system to meet the demand for rapid detection of fault points and then timely repair of equipment faults. The fault diagnosis of the field system is proved to have good accuracy by means of actual debugging summary or artificially created fault experiments.

The on-site human-machine interaction fault diagnosis phenomenon is shown in Fig 6. After the rapier loom has been running for a short period of time, the servo drive is artificially

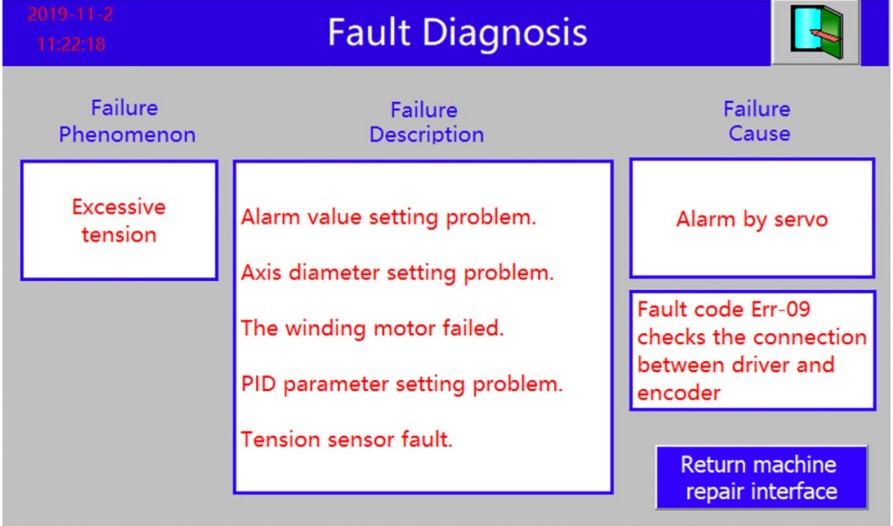

**Fig 6. Field human-computer interaction fault diagnosis effect.**

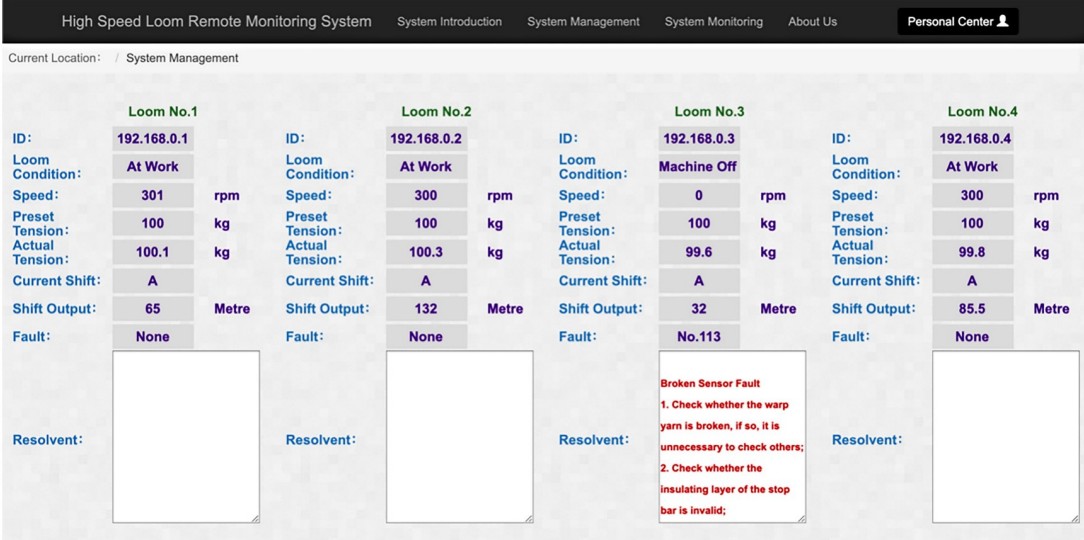

**Fig 7. Remote fault diagnosis effect of loom.**

disconnected from the encoder line end, thus preventing the servo motor from working properly. After the line end is disconnected the HCI field system issues a fault alarm to the control system to signal an emergency stop. A review of the fault alarm log revealed that the fabric tension was too high, and the diagnosis was derived from the reasoning of the fault diagnosis function for each detected parameter. The fault description introduces the possible causes of this fault type. Improper setting of the alarm value interval, wrong setting of the warp beam diameter, failure of the warp feeding and winding motor, wrong setting of the PID tension control algorithm parameters and failure of the tension sensor are all able to cause the fault phenomenon of excessive tension. According to the on-site system diagnosis of the cause of the fault, a fault was found in the servo motor drive, indicating a fault code Err-09 indicating a problem with the connection between the drive and the encoder. Field system fault diagnosis results are consistent with the servo drive alarm, and the fault phenomenon is explained to provide maintenance advice.

The effect of remote loom fault diagnosis is shown in Fig 7 loom No. 3 with ID 192.168.0.3 has a fault of weft breakage and the data of weft breakage sensor is abnormal. the fault diagnosis system calls the operation parameter of loom No. 3 and diagnoses the cause of the fault. It is judged that the fault situation of loom No. 3 is the same as the fault situation of training sample No. 113, and the weft break sensor is faulty. The fault diagnosis system gives the solution according to the fault situation of training sample 113: 1. check whether the warp yarn is broken, if so, no other checks are necessary; 2. check whether the insulation layer of the warp stopper is not working. The system monitors several looms remotely at the same time. Once the loom stops working with a fault, the cloud-based fault diagnosis system immediately performs a fault diagnosis and gives a solution to troubleshoot the fault based on the knowledge, so that maintenance personnel can quickly troubleshoot the loom based on the solution.

The fault diagnosis system in the cloud can be trained on fault data using algorithms of rough sets and Bayesian networks because it is deployed in the cloud. Therefore, more fault situations can be diagnosed compared to the expert system in the field. The expert system in the field handles common fault problems and reduces the computational pressure in the cloud.

Other fault problems are handled by the fault diagnosis system in the cloud. When other fault problems are diagnosed incorrectly, the fault data obtained from the cloud is used to continue the training and improve the correct rate of fault diagnosis.

## Conclusion

In this paper, we study a collaborative cloud-edge rapier loom fault system that performs edge computing to troubleshoot common faults at the industrial site and other fault diagnostics in the cloud. Edge computing shares the computational load of the cloud to handle common fault situations. When edge computing cannot effectively diagnose faults, optimized fault diagnosis algorithms are trained in the cloud to handle fault problems using the cloud-based fault diagnosis system. The expert system built in the field is able to handle common fault problems of rapier looms. The cloud-based fault diagnosis method for rapier looms based on rough sets and Bayesian networks can handle most other fault problems. The fault diagnosis method in the cloud can be continuously optimized by using the fault data in the cloud, and the expert system in the field applies the idea of edge computing to improve the real-time and reliability of the whole system. This system provides a technical example for the weaving machine fault diagnosis system, and optimizes the original fault diagnosis system according to the characteristics of the fault diagnosis object by using the fault diagnosis method of cloud-edge collaboration. The computing resources of the loom HMI system are used for edge computing to deal with common fault problems. It avoids the situation that the computational task of fault diagnosis in the cloud is overloaded because of the simultaneous occurrence of faults in several looms, and reduces the pressure on the cloud computing.

## Supporting information

**S1 File.**
(DOCX)

## Acknowledgments

Thanks for the support of equipment and experimental site provided by Jiangsu keread Intelligent Control Automation Technology Co., Ltd.

## Author Contributions

**Conceptualization:** Yanjun Xiao, Kuan Wang, Feng Wan.

**Formal analysis:** Kuan Wang.

**Funding acquisition:** Yanjun Xiao.

**Investigation:** Kuan Wang.

**Methodology:** Yanjun Xiao, Feng Wan.

**Project administration:** Yanjun Xiao.

**Software:** Weiling Liu.

**Supervision:** Weiling Liu.

**Validation:** Kai Peng.

**Visualization:** Kai Peng.

**Writing – original draft:** Weiling Liu.

**Writing – review & editing:** Feng Wan.

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
