## [Decision Letter · Decision Letter 0]

17 Jun 2021

PONE-D-21-16599

Research on rapier loom fault system based on cloud-side collaboration

PLOS ONE

Dear Dr. Wan,

Thank you for submitting your manuscript to PLOS ONE. After careful consideration, we feel that it has merit but does not fully meet PLOS ONE’s publication criteria as it currently stands. Therefore, we invite you to submit a revised version of the manuscript that addresses the points raised during the review process.

ACADEMIC EDITOR:

Based on the comments received from the reviewers and my own observation, I recommend major revisions for the article.

We look forward to receiving your revised manuscript.

Kind regards,

Thippa Reddy Gadekallu

Academic Editor

PLOS ONE

Journal Requirements:

 [No].

Additional Editor Comments (if provided):

Reviewers' comments:

Reviewer's Responses to Questions

**Comments to the Author**

1. Does the manuscript provide a valid rationale for the proposed study, with clearly identified and justified research questions?

Reviewer #1: Yes

Reviewer #2: Yes

2. Is the protocol technically sound and planned in a manner that will lead to a meaningful outcome and allow testing the stated hypotheses?

Reviewer #1: Yes

Reviewer #2: Yes

3. Is the methodology feasible and described in sufficient detail to allow the work to be replicable?

Reviewer #1: Yes

Reviewer #2: Yes

4. Have the authors described where all data underlying the findings will be made available when the study is complete?

Reviewer #1: Yes

Reviewer #2: Yes

5. Is the manuscript presented in an intelligible fashion and written in standard English?

Reviewer #1: Yes

Reviewer #2: Yes

6. Review Comments to the Author

You may also provide optional suggestions and comments to authors that they might find helpful in planning their study.

Reviewer #1: comment-1: According to my understanding, the authors classified the electrical faults of rapier loom into common faults and other faults according to the frequency of occurrence by analyzing process and fault characteristics of rapier loom. Then an expert system is built in the field for edge computing based on knowledge fault diagnosis experience to diagnose common loom faults and reduce the computing pressure in the cloud.

comment-2: The authors claims that this study also "reduce the computing pressure in the cloud", but it is only written in abstract. The authors should also explain this point in intro, proposed and conclusion sections.

comment-3: Paper is well written. Authors should add limitations of the proposed work.

Comment-4: The table-1 can be written in more organized form such that to clear understanding about parameters.

comment-5: The authors should add some more words about HMI.

comment-6: The quality of the figures can be improved more especially the figure-6 and 7. Figures should be eye-catching not blur. It will enhance the interest of the reader.

commnet-7: The authors should add more clear detail in a paragraph on figure-4 by making comparison of used algorithms.

commetn-8: The classification results should also be present in abstract in 1 or 2 sentences.

comment-9: The summary at the end of the literature review should be focused on the limitations of related work.

commnet-10: Authors should explain at least the features of the dataset in a table.

commnet-11: Authors should add the most recent references related to cloud computing:

(1) M. Shabbir et al., "Enhancing Security of Health Information Using Modular Encryption Standard in Mobile Cloud Computing," in IEEE Access, vol. 9, pp. 8820-8834, 2021, doi: 10.1109/ACCESS.2021.3049564.

(2) A. R. Javed, M. Usman, S. U. Rehman, M. U. Khan and M. S. Haghighi, "Anomaly Detection in Automated Vehicles Using Multistage Attention-Based Convolutional Neural Network," in IEEE Transactions on Intelligent Transportation Systems, doi: 10.1109/TITS.2020.3025875.

(3) Swarna Priya R.M., Sweta Bhattacharya, Praveen Kumar Reddy Maddikunta, Siva Rama Krishnan Somayaji, Kuruva Lakshmanna, Rajesh Kaluri, Aseel Hussien, Thippa Reddy Gadekallu,

Load balancing of energy cloud using wind driven and firefly algorithms in internet of everything,

Journal of Parallel and Distributed Computing,

Volume 142,

2020,

Pages 16-26,

ISSN 0743-7315,

https://doi.org/10.1016/j.jpdc.2020.02.010.

(4) Thar Baker, Bandar Aldawsari, Muhammad Asim, Hissam Tawfik, Zakaria Maamar, Rajkumar Buyya,

Cloud-SEnergy: A bin-packing based multi-cloud service broker for energy efficient composition and execution of data-intensive applications,

Sustainable Computing: Informatics and Systems,

Volume 19,

2018,

Pages 242-252,

ISSN 2210-5379,

https://doi.org/10.1016/j.suscom.2018.05.011.

(5) M. Asim, Y. Wang, K. Wang and P. -Q. Huang, "A Review on Computational Intelligence Techniques in Cloud and Edge Computing," in IEEE Transactions on Emerging Topics in Computational Intelligence, vol. 4, no. 6, pp. 742-763, Dec. 2020, doi: 10.1109/TETCI.2020.3007905.

Reviewer #2: • In Introduction section, the drawbacks of each conventional technique should be described clearly.

• Introduction needs to explain the main contributions of the work more clearly.

• The authors should emphasize the difference between other methods to clarify the position of this work further.

• The Wide ranges of applications need to be addressed in Introductions

• The quality of the figures can be improved, in terms of resolution and dimensions. The authors can add the refer the applications of cloud. Load balancing of energy cloud using wind driven and firefly algorithms in internet of everything. Providing diagnosis on diabetes using cloud computing environment to the people living in rural areas of India. A Blockchain Based Cloud Integrated IoT Architecture Using a Hybrid Design.

7. PLOS authors have the option to publish the peer review history of their article (what does this mean?). If published, this will include your full peer review and any attached files.

Reviewer #1: No

Reviewer #2: No

---

## [Author Response · Author response to Decision Letter 0]

10 Nov 2021

Reviewer #1: 

comment-1: According to my understanding, the authors classified the electrical faults of rapier loom into common faults and other faults according to the frequency of occurrence by analyzing process and fault characteristics of rapier loom. Then an expert system is built in the field for edge computing based on knowledge fault diagnosis experience to diagnose common loom faults and reduce the computing pressure in the cloud.

Reply-1: Thanks to the experts for their valuable comments on this article. You have summarized it very well, maybe I need to add some content to facilitate other readers to understand the work better.

Revision-1: Add research context and research questions related to reducing stress in cloud computing to highlight the purpose of the study.

comment-2: The authors claims that this study also "reduce the computing pressure in the cloud", but it is only written in abstract. The authors should also explain this point in intro, proposed and conclusion sections.

Reply-2: Common weaving machine faults can also be diagnosed by cloud computing, however, due to the high frequency of occurrence, knowledge for fault diagnosis can be extracted from the fault data samples. Using this knowledge loom equipment can be used for fault diagnosis using the available computing and storage resources. The existing loom fault diagnosis methods are in the cloud platform for these fault situations also using artificial intelligence algorithms. Due to the high frequency of these loom failures, the existing loom fault diagnosis methods can cause a waste of computing resources in the cloud platform. Under the future development trend of data explosion, significant increase in the number of equipment access, and shortage of data computing resources in the cloud platform, the existing loom fault diagnosis method needs to be improved to reduce the pressure on the cloud computing.

Revision-2: In the introduction section it is explained that this fault diagnosis method reduces the stress of cloud computing compared to the original method. This is further explained in the main content by describing this method of loom fault diagnosis and analyzing the workload of cloud computing using this method and the original method, which is again emphasized in the conclusion section.

comment-3: Paper is well written. Authors should add limitations of the proposed work.

Reply-3: The contribution of the article may need to be more explicit to illustrate the limitations of the work.

Revision-3: Added the necessary conditions for the work in the introduction, especially the problems targeted by the loom fault diagnosis system of Cloud Edge Collaboration.

comment-4: The table-1 can be written in more organized form such that to clear understanding about parameters.

Reply-4: I use table 1 to express the common loom faults and their solutions, but it may seem too bloated to clearly express the parameters.

Revision-4: In order to better express, I split table 1 into two tables. Table 2 uses the number to identify the troubleshooting method of loom, and table 3 uses the number of Table 1 to express the common loom faults and their solutions, so the content is more compact.

comment-5: The authors should add some more words about HMI.

Reply-5: The article may lack a more specific design of the weaving machine human-machine interface, which has been added in the main content, and is not developed in detail since the human-machine interface is not the focus of the discussion.

Revision-5: In the section of the article where the main weaving machine processes are described, the design of the human-machine interface of the weaving machine is explained.

comment-6: The quality of the figures can be improved more especially the figure-6 and 7. Figures should be eye-catching not blur. It will enhance the interest of the reader.

Reply-6: The definition of the figures are not enough, because the figures will become blurred when enlarged.

Revision-6: The size, resolution and sharpness of all figures have been adjusted.

commnet-7: The authors should add more clear detail in a paragraph on figure-4 by making comparison of used algorithms.

Reply-7: The paragraphs in Figure-4 mainly describe the diagnostic effect of the loom fault diagnosis method based on rough sets and Bayesian networks, and the description of other algorithms is rather vague and has been modified.

Revision-7: Added a detailed description of the diagnostic effects of other algorithms, and more detailed description of the correctness and volatility of the diagnostic results.

commetn-8: The classification results should also be present in abstract in 1 or 2 sentences.

Reply-8: Detailed data on the results of the fault classification cannot be presented for commercial reasons, but a general description of the results of the classification can be given and corresponding modifications have been made.

Revision-8: After the fault analysis of the loom, the results of classifying the loom faults into common faults and other faults are described.

comment-9: The summary at the end of the literature review should be focused on the limitations of related work.

Reply-9: At the end of the literature review, the limitations of the related work are summarized.

The literature review only mentions that the fault diagnosis method cannot be applied to the loom and lacks a summary of the limitations of the work related to the loom, which has been made to correspond to the modifications.

Revision-9: The limitations of the related work are explained at the end of the literature review to address the issues studied in this work.

commnet-10: Authors should explain at least the features of the dataset in a table.

Reply-10: In accordance with the formatting requirements of the journal, data preparation should be carried out in the main content, suitable for the insertion of a table explaining the characteristics of the data set, relevant modifications have been made.

Revision-10: Added a paragraph and a table to represent the characteristics of the dataset for troubleshooting.

commnet-11: Authors should add the most recent references related to cloud computing:

(1) M. Shabbir et al., "Enhancing Security of Health Information Using Modular Encryption Standard in Mobile Cloud Computing," in IEEE Access, vol. 9, pp. 8820-8834, 2021, doi: 10.1109/ACCESS.2021.3049564.

(2) A. R. Javed, M. Usman, S. U. Rehman, M. U. Khan and M. S. Haghighi, "Anomaly Detection in Automated Vehicles Using Multistage Attention-Based Convolutional Neural Network," in IEEE Transactions on Intelligent Transportation Systems, doi: 10.1109/TITS.2020.3025875.

(3) Swarna Priya R.M., Sweta Bhattacharya, Praveen Kumar Reddy Maddikunta, Siva Rama Krishnan Somayaji, Kuruva Lakshmanna, Rajesh Kaluri, Aseel Hussien, Thippa Reddy Gadekallu,

Load balancing of energy cloud using wind driven and firefly algorithms in internet of everything,

Journal of Parallel and Distributed Computing,

Volume 142,

2020,

Pages 16-26,

ISSN 0743-7315,

https://doi.org/10.1016/j.jpdc.2020.02.010.

(4) Thar Baker, Bandar Aldawsari, Muhammad Asim, Hissam Tawfik, Zakaria Maamar, Rajkumar Buyya,

Cloud-SEnergy: A bin-packing based multi-cloud service broker for energy efficient composition and execution of data-intensive applications,

Sustainable Computing: Informatics and Systems,

Volume 19,

2018,

Pages 242-252,

ISSN 2210-5379,

https://doi.org/10.1016/j.suscom.2018.05.011.

(5) M. Asim, Y. Wang, K. Wang and P. -Q. Huang, "A Review on Computational Intelligence Techniques in Cloud and Edge Computing," in IEEE Transactions on Emerging Topics in Computational Intelligence, vol. 4, no. 6, pp. 742-763, Dec. 2020, doi: 10.1109/TETCI.2020.3007905.

Reply-11: In order to illustrate the work of this paper, the latest work on cloud computing needs to be introduced in the introduction.

Revision-11: Relevant references have been cited in the article to illustrate the research related to cloud computing.

Reviewer #2:

• In Introduction section, the drawbacks of each conventional technique should be described clearly.

Reply-1: We thank the experts for their valuable comments on this article. The article does not highlight the traditional technology enough when introducing it, and the description of the disadvantages is not detailed enough. Changes have been made to this section.

Revision-1: For each traditional technology introduce the contribution with a description of its shortcomings, rather than a general description at the end.

• Introduction needs to explain the main contributions of the work more clearly.

Reply-2: The article is not sufficiently detailed in its description of the background and problems of the study, and the significance of the work is not adequately explained. The main contribution of the work has been highlighted by the description of the research questions and the significance of the work.

Revision-2: Provide more detail on the background and problems of the study and more detail on the purpose of the study. Highlight the study's best features and main contributions.

• The authors should emphasize the difference between other methods to clarify the position of this work further.

Reply-3: The article lacks the highlighting of the characteristics and advantages of the study in comparing other methods, especially the differences in research questions and research effects. Revisions have been made.

Revision-3: A summary of other methods outlines issues not covered by other methods and describes the research conducted by the proposed method in response to this method. The characteristics of the research and the position of the work in this paper are emphasized in the context of the previous research.

• The Wide ranges of applications need to be addressed in Introductions.

Reply-4: The article does not discuss the value of this approach to be borrowed in other application scenarios. In particular, the theoretical reusability of this approach in other application scenarios. The concept of dividing the computational tasks for application specific needs and distributing them rationally in the field and in the cloud should be discussed.

Revision-4: The boundary conditions of whether this approach of cloud edge synergy can be used and studied in other scenarios of application requirements are discussed.

• The quality of the figures can be improved, in terms of resolution and dimensions. The authors can add the refer the applications of cloud. Load balancing of energy cloud using wind driven and firefly algorithms in internet of everything. Providing diagnosis on diabetes using cloud computing environment to the people living in rural areas of India. A Blockchain Based Cloud Integrated IoT Architecture Using a Hybrid Design.

Reply-5: The definition of the figures are not enough, because the figures will become blurred when enlarged.In order to illustrate the work of this paper, reference literature related to cloud computing needs to be added in the introduction.

Revision-5: The size, resolution and sharpness of all figures have been adjusted.Relevant references have been cited in the article to illustrate the research related to cloud computing.

---

## [Decision Letter · Decision Letter 1]

19 Nov 2021

Research on rapier loom fault system based on cloud-side collaboration

PONE-D-21-16599R1

Dear Dr. Wan,

We’re pleased to inform you that your manuscript has been judged scientifically suitable for publication and will be formally accepted for publication once it meets all outstanding technical requirements.

Kind regards,

Thippa Reddy Gadekallu

Academic Editor

PLOS ONE

Additional Editor Comments (optional):

Reviewers' comments:

Reviewer's Responses to Questions

**Comments to the Author**

1. Does the manuscript provide a valid rationale for the proposed study, with clearly identified and justified research questions?

Reviewer #3: Yes

2. Is the protocol technically sound and planned in a manner that will lead to a meaningful outcome and allow testing the stated hypotheses?

Reviewer #3: Yes

3. Is the methodology feasible and described in sufficient detail to allow the work to be replicable?

Reviewer #3: Yes

4. Have the authors described where all data underlying the findings will be made available when the study is complete?

Reviewer #3: Yes

5. Is the manuscript presented in an intelligible fashion and written in standard English?

Reviewer #3: Yes

6. Review Comments to the Author

You may also provide optional suggestions and comments to authors that they might find helpful in planning their study.

Reviewer #3: The contribution is exciting and considerably novel. The paper is well written and starts with influencing motivation, and is reflected throughout the paper. I would like to accept this paper.

7. PLOS authors have the option to publish the peer review history of their article (what does this mean?). If published, this will include your full peer review and any attached files.

Reviewer #3: No

---

## [Editor Report · Acceptance letter]

10 Dec 2021

PONE-D-21-16599R1 

Research on rapier loom fault system based on cloud-side collaboration 

Dear Dr. Wan:

I'm pleased to inform you that your manuscript has been deemed suitable for publication in PLOS ONE. Congratulations! Your manuscript is now with our production department. 

Kind regards, 

on behalf of

Dr. Thippa Reddy Gadekallu 

Academic Editor

PLOS ONE